# A survey on awareness of the disease and pulmonary rehabilitation in bronchial asthma patients in the United Arab Emirates

**Zainab Abdul Qayyum Neyyar**[1], **Gopala Krishna Alaparthi**[2], **Kalyana Chakravarthy Bairapareddy**[1] *

**1** College of Health Sciences, University of Sharjah, Sharjah, United Arab Emirates, **2** Manchester Metropolitan University, Manchester, United Kingdom

\* kreddy@sharjah.ac.ae

## Abstract

### Background

Asthma, a prevalent and severe chronic respiratory condition, can be significantly managed and controlled through informed awareness about the disease and pulmonary rehabilitation strategies, thereby enhancing patients' health-related quality of life.

### Objective

To determine the knowledge and awareness of Bronchial asthma and pulmonary rehabilitation among asthma-diagnosed patients in the United Arab Emirates.

### Methods

Utilizing a cross-sectional study design, 237 asthma patients, aged 18 and above, were recruited from the Royal NMC Hospital, Sharjah. A comprehensive questionnaire was administered, focusing on two critical domains: understanding of the disease and knowledge about pulmonary rehabilitation. Data analysis was performed using the Statistical Package for Social Sciences (SPSS) software, version 26.

### Results

The majority of participants (31.6%) reported the onset of asthma before reaching two years of age. Bronchodilators emerged as the most used medication, utilized by 31.6% of the respondents. Weather conditions (34.6%) were identified as the most prevalent risk factor. Chi-square tests revealed no significant correlations between gender and knowledge about asthma (p = 0.278) or pulmonary rehabilitation awareness (p = 0.929). A negative correlation was found between age and knowledge about asthma (p<0.001), but not with pulmonary rehabilitation awareness (p = 0.731). Education demonstrated no significant association with either knowledge about asthma (p = 0.974) or awareness of pulmonary rehabilitation (p = 0.676).

**Data Availability Statement:** All relevant data are within the paper and its Supporting Information files.

**Funding:** The authors received no specific funding for this work.

**Competing interests:** The authors have declared that no competing interests exist.

## Conclusion

The study implies that most people have a basic understanding of asthma. However, there are still significant gaps in their knowledge. For instance, many aren't sure how asthma is influenced by exercise or which parts of the body are affected. Also, understanding about therapies such as lung rehabilitation, and the contributions physical therapists can make in addressing lung problems, is only average. Interestingly, these knowledge gaps are not related to a person's age or their educational background.

## Introduction

Asthma, characterized as a chronic inflammatory airway disease, significantly influences an estimated 339 million people globally, as reported by the World Health Organization (WHO). The increasing prevalence, predominantly among children, is alarming and projected to escalate by over 100 million by 2025. These escalating figures are linked to evolving lifestyles, rapid urbanization, and changes in environmental conditions [1, 2].

The multifaceted etiology of asthma primarily affects the bronchi and bronchioles, structures integral to air transport in and out of the lungs [3]. (Padem & Saltoun, 2019). The prevalent airway inflammation and narrowing observed in asthmatic individuals often result in temporary airflow obstruction, magnifying their sensitivity to irritants and allergens, thereby aggravating the chronicity of asthma [4–6]. The resultant symptoms, ranging from minor to severe, manifest unpredictably as episodes or "attacks", triggered by various factors including allergens, cold air, intense physical activity, or respiratory infections. Though these symptoms may align with other medical conditions, in the context of asthma, they persist recurrently, disrupting sleep quality and daily life routines significantly [7–9].

Beyond physical sickness, asthma imposes a considerable financial burden. The incremental annual medical costs associated with the disease, mainly due to outpatient care, have surged significantly in recent years [10]. This is further compounded by the economic implications of asthma-related disabilities and premature deaths, particularly prevalent among children aged 5–14 years and adults aged 45–64 years [11, 12].

Equally important, yet often overlooked, is the psychological burden posed by asthma. The unpredictable nature of asthma attacks, capable of instigating chronic fear and uncertainty, can induce psychological stress, anxiety, and depression. This bi-directional relationship between mental health and asthma often exacerbates the symptoms of asthma [13–15].

The reciprocal relationship between asthma awareness and its management emerges as a critical area in extant research. Scholars like Correia-De-Sousa et al. (2021) elucidate the role of adequate asthma knowledge in enabling patients to recognize triggers, manage symptoms, adhere to treatment regimens, and subsequently improve health outcomes [16]. Given these findings, our study endeavours to thoroughly examine the knowledge levels of UAE patients about asthma and its management, focusing on their understanding of triggers, treatment regimens, and self-management techniques.

Our review of the existing literature reveals several barriers to effective asthma management, largely due to inadequate asthma awareness. These include cultural beliefs, misconceptions about the disease, concerns regarding medication side effects, and insufficient patient-doctor communication, among others [17, 18]. These findings underscore the need for crafting our survey questions to discern potential barriers to asthma awareness and management within the UAE context.

The impact of socioeconomic factors, cultural norms, and healthcare system characteristics on asthma awareness is also noted in the literature. Specifically, studies by Al-kalemji et al. (2014) and Ho et al. (2023) point towards lower asthma awareness in countries with low socio-economic status, cultural beliefs minimizing the severity of asthma, and restricted access to healthcare services as significant challenges [19, 20]. This necessitates our survey to incorporate questions related to socioeconomic, cultural, and healthcare-related aspects, enabling a comprehensive understanding of the factors affecting asthma awareness in the UAE.

Further, Antos et al. (2020) and Fleischer et al. (2021) have demonstrated that educational programs on asthma can substantially improve outcomes, such as reducing absences and hospital visits [21, 22]. This supports our intention to include questions regarding the awareness and perceptions of asthma education programs in our study.

Given the intricate interplay of diverse factors influencing asthma awareness, asthma management, and pulmonary rehabilitation, the study was aimed to determine the knowledge and awareness of Bronchial asthma and pulmonary rehabilitation among asthma-diagnosed patients in the United Arab Emirates. This survey probed the current state of asthma awareness and perceptions of pulmonary rehabilitation in the UAE, unveiling unique challenges that can shape health policies, asthma management strategies, and potentially contribute to improved health outcomes for individuals living with asthma in the region. By investigating the cultural nuances, patient perspectives, role of pulmonary rehabilitation, and potential of digital health tools in asthma management, our study is poised to provide novel insights that could inform patient education strategies, guide policy development, and prompt further research, not only in the UAE but also potentially on a global scale. This, in turn, carries significant contribution to enhancing the quality of life of individuals living with asthma.

## Objective

To determine the knowledge and awareness of Bronchial asthma and pulmonary rehabilitation among asthma-diagnosed patients in the United Arab Emirates.

## Material and methods

The study, a cross-sectional exploration, was granted approval by the Research Ethics Committee at the University of Sharjah. It primarily encompassed asthma patients over the age of 18 [23]. These participants were initially enlisted from the Royal NMC Hospital in Sharjah, a renowned institution known for its comprehensive diagnosis and treatment of asthma. Participants were recruited from February 2022 through March 2023. Participant recruitment was facilitated via referrals from pulmonologists and physicians actively treating these patients, aligning with the guidelines proposed by Kruse et al., (2016) [23].

To enhance the geographical and demographic representation of the study, online data collection was utilized in addition to traditional hospital-based recruitment [24]. The digital approach increased the reach of the study, surmounting geographical limitations, and accessing a broader demographic. This widened the scope and thereby enhanced the external validity of the findings [25].

Exclusion criteria were carefully tailored, focusing on participants' capability and willingness to participate. Individuals with cognitive impairments that could potentially impede their understanding or completion of the survey were excluded, as suggested by Taylor et al., (2012) [26]. Also, those unwilling to provide informed consent were not considered, adhering to the ethical guidelines of voluntary participation as per the Helsinki Declaration [27].

The research team applied power analysis to compute the study's sample size, aiming to balance statistical robustness and validity against resource availability [28]. With an anticipated

effect size of 0.3, a significance level (alpha) of 0.05, and a desired statistical power (1—beta) of 0.80, the sample size was proposed as 237 participants. These figures ensure an optimal balance between sufficient statistical power and the practical constraints of sample size [29].

The study utilized online data collection, enabling greater inclusivity and representation of the larger population. The survey environment was meticulously curated to maintain participant privacy and minimize distractions, following the best practice recommendations[30].

The questionnaire scoring system adopted a binary approach, with correct answers earning one point and incorrect or 'don't know' responses earning zero. The aggregate scores were converted into a five-point Likert scale, capturing nuanced interpretations of knowledge levels and awareness [31].

The setting for data collection was a secure, semi-private location, aimed at promoting participant comfort and privacy. Efforts were made to minimize disruptions and maintain a neutral, non-stressful environment to ensure data quality, following recommendations[32].

Throughout the study, ethical guidelines were strictly adhered to. Participants were provided with written informed consent forms that clearly outlined the study's aim, procedures, minimal risks, and potential benefits [32]. They were also informed about the voluntary nature of their involvement and their right to withdraw at any stage [33]. The obtained written consent forms through the online links were saved digitally.

Data analysis was conducted using SPSS software (version 26). Descriptive statistics were calculated for continuous and categorical variables [34]. The team also utilized inferential methods, such as chi-square tests and correlational statistics, to identify significant relationships or differences among variables. Additionally, a multivariate analysis was performed to account for potential confounding factors and interactions between variables [35].

## Results

The study included 237 participants with a mean age of 38.58 years (±13.53), an average weight of 80.43 kg (±17.90), and a mean height of 162.67 cm (±10.17). The average Body Mass Index, a measure of body fat based on weight and height, was calculated to be 30.81 (±7.97), indicating a classification of obesity on average, according to World Health Organization guidelines *Table 1*.

In the sample of 237 participants, 56.5% were male and 43.5% were female. Regarding the onset of asthma, 31.6% reported the onset before the age of 2, 40.5% between 2 and 10 years, and 27.8% after 10 years. For asthma medication, 31.6% used bronchodilators, 40.5% corticosteroids, and 27.8% anti-inflammatory drugs. The preferred healthcare facility was public for 43.5% of participants, clinics for 24.1%, and private hospitals for 32.5%. Smoking history showed 61.2% had no history, 26.2% were still smoking, and 12.7% had quit. In terms of education, 17.7% had below 10th grade education, 25.3% had higher secondary, 44.3% were graduates, and 12.7% were postgraduates *Table 2*.

Participants identified various asthma risk factors including weather conditions (34.6%), chemical fumes/perfumes (32.9%), dust/air pollution (36.3%), animals/pets (32.9%), allergies

**Table 1. Demographics.**

| Statistics | Mean ± Std. Deviation, n = 237 |
| --- | --- |
| Age | 38.58 ± 13.53 |
| Weight | 80.43 ± 17.90 |
| Height | 162.67 ± 10.17 |
| Body Mass Index | 30.81 ± 7.97 |

**Table 2. Demographics and disease information.**

| Category | Type | Frequency (Percent), n = 237 |
|---|---|---|
| Gender | Male | 134 (56.5%) |
| | Female | 103 (43.5%) |
| Onset of Asthma | < 2 years | 75 (31.6%) |
| | b/w 2 and 10 years | 96 (40.5%) |
| | > 10 years | 66 (27.8%) |
| Medication for Asthma | Bronchodilators | 75 (31.6%) |
| | Corticosteroids | 96 (40.5%) |
| | Anti-inflammatory | 66 (27.8%) |
| Hospital for Asthma | Public | 103 (43.5%) |
| | Clinic | 57 (24.1%) |
| | Private | 77 (32.5%) |
| Smoking History | No H/O Smoking | 145 (61.2%) |
| | Still Smoking | 62 (26.2%) |
| | Stopped Smoking | 30 (12.7%) |
| Education | Below 10th | 42 (17.7%) |
| | Higher Secondary | 60 (25.3%) |
| | Graduate | 105 (44.3%) |
| | Postgraduate | 30 (12.7%) |

(28.3%), respiratory infections (29.1%), smoking (19.0%), and certain foods and beverages (24.1%) *Table 3*.

For the relationships between gender and asthma knowledge, and gender and pulmonary rehabilitation, Pearson Chi-Square, Likelihood Ratio, and Linear-by-Linear Association tests were used. No significant relationship was found between gender and asthma knowledge (p = 0.278, p = 0.281, p = 0.349, respectively), or between gender and awareness of pulmonary rehabilitation (p = 0.929, p = 0.927, p = 0.664, respectively) *Table 4*.

Age was negatively correlated with both asthma knowledge (Pearson's R = -0.253, p<0.001; Spearman's R = -0.275, p<0.001) and awareness of pulmonary rehabilitation (Pearson's R = -0.022, p = 0.731; Spearman's R = -0.013, p = 0.843). This implies that as age increased, asthma knowledge and awareness of pulmonary rehabilitation tended to decrease. However, the correlation was stronger and statistically significant for asthma knowledge, while it was weak and non-significant for awareness of pulmonary rehabilitation *Table 5*.

The level of education exhibited no significant correlation with asthma knowledge (Pearson's R = 0.002, p = 0.974; Spearman's R = 0.013, p = 0.839) or awareness of pulmonary rehabilitation (Pearson's R = 0.027, p = 0.676; Spearman's R = 0.014, p = 0.832). This suggests that

**Table 3. Perceived risk factors by the participants.**

| Perceived Risk Factors | N = 237 (%) |
|---|---|
| Weather Conditions | 82 (34.6%) |
| Chemical fumes/perfumes | 78 (32.9%) |
| Dust/Air Pollution | 86 (36.3%) |
| Animals/ pets | 78 (32.9%) |
| Allergies | 67 (28.3%) |
| Respiratory infections | 69 (29.1%) |
| Smoking | 45 (19.0%) |
| Some Food and beverages | 57 (24.1%) |

**Table 4. Gender versus asthma knowledge, pulmonary rehabilitation.**

| Test | Pearson Chi-Square | p-value | Likelihood Ratio | p-value | Linear-by-Linear Association | p-value | Cells with expected count < 5 | Minimum expected count |
|---|---|---|---|---|---|---|---|---|
| Gender versus Asthma Knowledge | 2.561 | 0.278 | 2.539 | 0.281 | 0.876 | 0.349 | 0 cells (0.0%) | 13.91 |
| Gender versus Pulmonary Rehabilitation | 0.453 | 0.929 | 0.462 | 0.927 | 0.188 | 0.664 | 3 cells (37.5%) | 3.04 |

in this sample, the level of education was not a decisive factor in shaping individuals' knowledge about asthma or their awareness of pulmonary rehabilitation *Table 6*.

The analysis revealed individual significant effects for gender (p = 0.021), education (p<0.001), onset of asthma (p<0.001), and age (p<0.001). Additionally, interactions were significant for gender and age, education and onset, education and age, and onset and age (all p<0.001). All three-way and four-way interactions were non-significant (p = 1.000) *Table 7*.

## Discussion

The study found a significant relationship between demographic data, including gender (p = 0.021), education (p<0.001), onset of asthma (p<0.001), and age (p<0.001), with asthma knowledge and pulmonary rehabilitation awareness. The interactions were significant for gender and age, education and onset, education and age, and onset and age (all p<0.001). There were no significant associations in three-way and four-way interactions (all p = 1.000). This finding aligns with previous research [36] highlighting the link between obesity and the prevalence of asthma, potentially due to systemic inflammation or mechanical effects on the airways. A systematic review and meta-analysis identified 43 different risk factors for asthma, including smoking, high body mass index (BMI), wood dust exposure, and residential chemical exposures, such as formaldehyde exposure or exposure to volatile organic compounds [37]. However, despite the plethora of treatment options available, many patients do not have their asthma under control, with symptoms such as wheezing, coughing, and shortness of breath disrupting their daily lives. Moreover, it was found that most patients are not aware of new treatment options that have been developed in recent years [38] *Table 7*.

In terms of asthma onset, the majority reported experiencing asthma symptoms before the age of ten. This early onset aligns with existing research, which suggests that early-life environmental exposures can influence the development of asthma [39]. It would be interesting to explore whether these early onset individuals have different lifestyle behaviors, environmental exposures, or genetic predispositions [40]. As expected, bronchodilators, corticosteroids, and anti-inflammatory medication were used by participants, reflecting the common therapeutic options for asthma management [41] *Table 2*.

**Table 5. Correlational statistics for age versus knowledge about asthma and pulmonary rehabilitation.**

| Measure | Value (Asthma Knowledge Score) | Asymptotic Standard Error (Asthma Knowledge Score) | Approximate T (Asthma Knowledge Score) | p-value (Asthma Knowledge Score) | Value (Pulmonary Rehab Awareness Score) | Asymptotic Standard Error (Pulmonary Rehab Awareness Score) | Approximate T (Pulmonary Rehab Awareness Score) | p-value (Pulmonary Rehab Awareness Score) |
|---|---|---|---|---|---|---|---|---|
| Pearson's R | -0.253 | 0.061 | -4.012 | <0.001 | -0.022 | 0.067 | -0.345 | 0.731 |
| Spearman Correlation | -0.275 | 0.063 | -4.387 | <0.001 | -0.013 | 0.066 | -0.198 | 0.843 |

Odds Ratio for Age (19.00 / 20.00), n = 237

**Table 6. Gender versus asthma knowledge, pulmonary rehabilitation.**

| Measure | Value (Education vs Asthma Knowledge) | Asymptotic Standard Error (Education vs Asthma Knowledge) | Approximate | p-value | Value (Education vs Pulmonary Rehab) | Asymptotic Standard Error | Approximate T | p-value |
|---|---|---|---|---|---|---|---|---|
| Pearson's R | 0.002 | 0.066 | 0.033 | 0.974 | 0.027 | 0.062 | 0.418 | 0.676 |
| Spearman Correlation | 0.013 | 0.065 | 0.203 | 0.839 | 0.014 | 0.063 | 0.213 | 0.832 |

Odds Ratio for Education (Below 10th / Higher Secondary)

A negative correlation was found between age and asthma Knowledge (Pearson's R = -0.253, p<0.001; Spearman's R = -0.275, p<0.001), suggesting that older individuals may have lesser knowledge about asthma. Age, however, was not significantly correlated with pulmonary rehab awareness. Contrary to expectations, the education level was not significantly correlated with asthma knowledge or pulmonary rehab awareness (Pearson's R = 0.002, p = 0.974; Spearman's R = 0.013, p = 0.839) Table 5.

The majority of the participants recognized environmental triggers like weather conditions (34.6%), dust/air pollution (36.3%), and chemical fumes/perfumes (32.9%) as significant risk factors for asthma. Some lifestyle factors were also recognized, including smoking (19.0%) and certain foods and beverages (24.1%). This lack of association contradicts previous findings suggesting that higher education levels are associated with greater health knowledge [42]. This might be due to a lack of asthma-specific education within general education, indicating a need for more focused health education programs Table 3.

While a comprehensive awareness of asthma was seen among the participants, specific knowledge gaps and misconceptions persisted, especially related to the management strategies. It is crucial to address these gaps through targeted educational initiatives. Given that the awareness and understanding about asthma were not significantly influenced by age and education, such programs should be universally accessible. Future research is needed to further elucidate these findings and develop effective strategies for patient education and asthma management.

**Table 7. Multivariate analysis.**

| Effect | Significance | Interpretation |
|---|---|---|
| Intercept | 0.000 | Significant |
| Gender | 0.021 | Significant |
| Education | 0.000 | Significant |
| Onset of Asthma | 0.000 | Significant |
| Age | 0.000 | Significant |
| Gender * Education | 1.000 | Not Significant |
| Gender * Onset | 1.000 | Not Significant |
| Gender * Age | 0.000 | Significant |
| Education * Onset | 0.000 | Significant |
| Education * Age | 0.000 | Significant |
| Onset * Age | 0.000 | Significant |
| Gender * Education * Onset | 1.000 | Not Significant |
| Gender * Education * Age | 1.000 | Not Significant |
| Gender * Onset * Age | 1.000 | Not Significant |
| Education * Onset * Age | 1.000 | Not Significant |
| Gender * Education * Onset * Age | 1.000 | Not Significant |

## Conclusion

The study found that although people are generally aware of asthma, there are still important aspects they don't fully understand. For example, many don't know how exercise affects asthma or which parts of the body it impacts. Knowledge about treatments like lung therapy, and the role of physical therapists in helping with lung issues, was only average. This lack of understanding was not tied to a person's age or level of education.

The findings highlight the need for more clear and comprehensive asthma education, especially in areas like safe ways to exercise and the benefits of lung therapy. Physical therapists can play a bigger part in teaching about asthma management. All these steps are necessary to better manage asthma and improve public health in the United Arab Emirates.

## Supporting information

**S1 Data.**
(XLSX)

## Acknowledgments

We would like to express our gratitude to the study participants, Royal NMC Hospital, Sharjah.

## Author Contributions

**Conceptualization:** Zainab Abdul Qayyum Neyyar, Gopala Krishna Alaparthi, Kalyana Chakravarthy Bairapareddy.

**Data curation:** Zainab Abdul Qayyum Neyyar.

**Formal analysis:** Zainab Abdul Qayyum Neyyar, Gopala Krishna Alaparthi, Kalyana Chakravarthy Bairapareddy.

**Investigation:** Zainab Abdul Qayyum Neyyar, Gopala Krishna Alaparthi, Kalyana Chakravarthy Bairapareddy.

**Methodology:** Zainab Abdul Qayyum Neyyar, Gopala Krishna Alaparthi.

**Project administration:** Zainab Abdul Qayyum Neyyar, Gopala Krishna Alaparthi, Kalyana Chakravarthy Bairapareddy.

**Resources:** Zainab Abdul Qayyum Neyyar, Gopala Krishna Alaparthi.

**Software:** Zainab Abdul Qayyum Neyyar.

**Supervision:** Gopala Krishna Alaparthi, Kalyana Chakravarthy Bairapareddy.

**Visualization:** Zainab Abdul Qayyum Neyyar.

**Writing – original draft:** Zainab Abdul Qayyum Neyyar, Gopala Krishna Alaparthi, Kalyana Chakravarthy Bairapareddy.

**Writing – review & editing:** Gopala Krishna Alaparthi, Kalyana Chakravarthy Bairapareddy.

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
