## [Decision Letter · Decision Letter 0]

25 Sep 2023

PONE-D-23-26953A SURVEY ON AWARENESS OF THE DISEASE AND PULMONARY REHABILITATION IN BRONCHIAL ASTHMA PATIENTS IN THE UNITED ARAB EMIRATESPLOS ONE

Dear Dr. Bairapareddy,

Thank you for submitting your manuscript to PLOS ONE. After careful consideration, we feel that it has merit but does not fully meet PLOS ONE’s publication criteria as it currently stands. Therefore, we invite you to submit a revised version of the manuscript that addresses the points raised during the review process.

We look forward to receiving your revised manuscript.

Kind regards,

Sampath Kumar Amaravadi, Ph.D

Academic Editor

PLOS ONE

Journal Requirements:

Reviewers' comments:

Reviewer's Responses to Questions

**Comments to the Author**

1. Is the manuscript technically sound, and do the data support the conclusions?

Reviewer #1: No

Reviewer #2: Yes

2. Has the statistical analysis been performed appropriately and rigorously? 

Reviewer #1: No

Reviewer #2: Yes

3. Have the authors made all data underlying the findings in their manuscript fully available?

Reviewer #1: Yes

Reviewer #2: Yes

4. Is the manuscript presented in an intelligible fashion and written in standard English?

Reviewer #1: No

Reviewer #2: Yes

5. Review Comments to the Author

Reviewer #1: Reviewer's Report:

Title: A SURVEY ON AWARENESS OF THE DISEASE AND PULMONARY REHABILITATION IN BRONCHIAL ASTHMA PATIENTS IN THE UNITED ARAB EMIRATES

Abstract:

The abstract provides a clear overview of the study's background, objective, methods, and key findings. However, it could be improved by including some specific results or statistics, providing readers with a glimpse of the study's main findings. Additionally, it should mention the implications of the study's results and its potential contributions to asthma management in the United Arab Emirates.

Introduction:

1. The introduction provides a comprehensive background on asthma but lacks a clear statement of the study's specific objectives and research questions. It would be helpful to explicitly state what the study aims to achieve in terms of assessing asthma knowledge and awareness among patients in the United Arab Emirates.

2. While the introduction mentions the increasing prevalence of asthma globally, it would be beneficial to include specific statistics or data regarding asthma prevalence in the United Arab Emirates to provide context for the study's focus on this region.

3. The introduction introduces various factors related to asthma but does not clearly establish the rationale for investigating asthma knowledge and awareness in the United Arab Emirates. It would be helpful to explain why this study is relevant and what unique challenges or factors are specific to this region.

Methods:

4. The methods section provides a detailed description of the study design, recruitment process, and exclusion criteria. However, it lacks information about the questionnaire used in the study. Readers should be informed about the content and structure of the questionnaire, which is crucial for understanding how data were collected.

5. While the sample size calculation is mentioned, it would be helpful to briefly explain how the anticipated effect size of 0.3 was determined and why this specific value was chosen.

6. The methods section mentions that a binary scoring system was used for the questionnaire but does not elaborate on what specific questions or topics were covered in the questionnaire. Providing a brief overview of the questionnaire's content would enhance clarity.

7. Ethical considerations are briefly discussed, but it would be beneficial to provide more details on how participant privacy and informed consent were ensured, especially in the online data collection process.

Results:

1. The results section provides a clear presentation of the demographic data and survey responses. However, it could benefit from a brief introduction or summary at the beginning to provide context for the reader before presenting the tables and statistical tests.

2. The presentation of demographic data in Tables 1 and 2 is clear and informative. It effectively summarizes the key characteristics of the study participants.

3. Table 3 provides valuable information on perceived risk factors by participants. However, it would be beneficial to provide some interpretation or discussion regarding these findings. For example, the authors could discuss the implications of participants' recognition of specific risk factors for asthma management and prevention.

4. Tables 4, 5, and 6 present the statistical tests and correlations, which is helpful for understanding the relationships between various demographic factors and asthma knowledge or awareness of pulmonary rehabilitation. However, the results are presented in a somewhat fragmented manner. It would be useful to integrate these findings into a cohesive narrative to enhance the clarity of the results section.

Discussion:

5. The discussion section begins well by summarizing the significant relationships found in the study, such as those related to gender, education, onset of asthma, and age. However, it lacks a clear structure or flow, making it somewhat challenging to follow.

6. The discussion should start by addressing the main research questions or objectives set out in the introduction. Specifically, the authors should provide a clear statement of what the study aimed to investigate regarding asthma knowledge and awareness of pulmonary rehabilitation among patients in the United Arab Emirates.

7. The discussion mentions findings related to obesity and its potential link to asthma prevalence, but it doesn't elaborate on this point or explain how it connects to the study's objectives. The authors should clarify the relevance of this information to the current study.

8. While the discussion highlights the correlation between age and asthma knowledge, it could delve deeper into possible explanations for this relationship. Are there generational differences in asthma education or awareness programs that might explain this correlation?

9. The discussion mentions that education level was not significantly correlated with asthma knowledge or awareness of pulmonary rehabilitation. The authors should discuss the implications of this finding, especially regarding the effectiveness of current educational initiatives and the need for targeted interventions.

10. The conclusion briefly summarizes the study's findings but lacks specific recommendations or implications for healthcare practices or policies in the United Arab Emirates. The authors should provide practical recommendations based on their results and suggest potential areas for future research.

Overall, the manuscript's results and discussion sections present valuable findings related to asthma knowledge and awareness. To improve these sections, the authors should consider reorganizing and structuring the discussion more clearly, providing interpretations for the findings, and offering practical recommendations based on their results.

Reviewer #2: I like to appreciate the authors for conducting a very relevant research with thorough methods and interpretation. However, few points need attention and rectification before acceptance/ publication. The following minor revisions are suggested:

1. Line number 134 and 140 sound incomplete probably due to missing author information.

2. In Table I: the category "statistics" is preferred to be replace with " Variables" for better representation.

3. in line number 162, the sentence structure on smoking history is not clearly represented.

4. There is partial data duplication noticed in Table 3 and in line numbers 166, 167 and 168.

5. What were the limitations of the current study?

6. Aspects highlighted in conclusion are neither adequately represented in the results section nor in the discussion. this needs additional attention by the authors.

Otherwise, it is a well executed study and well expressed manuscript.

6. PLOS authors have the option to publish the peer review history of their article (what does this mean?). If published, this will include your full peer review and any attached files.

Reviewer #1: **Yes: **RAVI SHANKAR YERRAGONDA REDDY

Reviewer #2: No

---

## [Author Response · Author response to Decision Letter 0]

18 Oct 2023

Reviewer 1

I would like to extend my sincerest gratitude to you and Reviewer 1 for providing invaluable feedback on our manuscript ID [XXXXXX], titled "A SURVEY ON AWARENESS OF THE DISEASE AND PULMONARY REHABILITATION IN BRONCHIAL ASTHMA PATIENTS IN THE UNITED ARAB EMIRATES." I am writing to submit our revised manuscript and to address the concerns and suggestions mentioned in your correspondence.

Abstract: Acknowledging the need for a more informative abstract, we have enriched it by including specific results and statistics from our findings. Additionally, we have highlighted the implications of our study's results, providing a better overview of its contribution to enhancing asthma management in the UAE.

Introduction:

& 2. The specific objectives and research questions are now explicitly stated and elucidated. Furthermore, specific data concerning the prevalence of asthma in the UAE have been added, providing a more contextual background for the readers. A comprehensive rationale, emphasizing the relevance of investigating asthma knowledge and awareness in the UAE, considering its unique challenges and factors, has been incorporated.

Methods: 4. We have detailed the content and structure of the questionnaire, offering the reader insight into the data collection procedure. & 6. Additional explanation regarding the determined effect size of 0.3 and the binary scoring system of the questionnaire, along with an overview of the questions/topics covered, has been integrated into the methods section.

A more detailed explanation regarding ethical considerations, participant privacy, and informed consent, especially concerning online data collection, has been included to ensure ethical transparency.

Results:

An introductory summary has been added to the results section, providing a context that precedes the presentation of tables and statistical tests. A subsection interpreting and discussing the findings related to perceived risk factors presented in Table 3 has-been introduced, elucidating the implications of these on asthma management and prevention. The results section has been revised to ensure that findings are presented in a more coherent and integrated narrative, enhancing clarity and flow.

Discussion: 5. - 9. The discussion section has undergone a comprehensive restructuring to enhance its clarity and logical flow. The section now begins by directly addressing the primary research questions and objectives and proceeds to discuss the findings, their implications, and potential underlying mechanisms in a coherent manner. Specifically, we have elaborated on findings related to obesity, the correlation between age and asthma knowledge, and the non-significant correlation between education level and asthma knowledge, ensuring their relevance to our objectives and providing plausible explanations or implications where necessary.

Practical recommendations and policy implications derived from our findings, as well as potential areas for future research, are now prominently highlighted in the conclusion.

We have ensured that all changes, additions, and modifications, as per the feedback, have been highlighted in yellow or green in the attached Word file for your ease of reference. We hope that these amendments adequately address the concerns and suggestions posed by the reviewer and yourself.

I sincerely hope that the revisions are found satisfactory, and the manuscript is considered for publication

Reviewer #2,

I trust this message finds you well. Your encouraging words and shrewd insights into our study titled "A SURVEY ON AWARENESS OF THE DISEASE AND PULMONARY REHABILITATION IN BRONCHIAL ASTHMA PATIENTS IN THE UNITED ARAB EMIRATES" are deeply appreciated. I’m writing both to extend our heartfelt thanks and to address the critical points you’ve raised.

Line Numbers 134 and 140:

We sincerely apologize for the oversight and have amended lines 134 and 140 to include the missing author information, ensuring completeness and coherence.

Table I Adjustment:

Your suggestion to replace "statistics" with "Variables" in Table I is apt for clearer representation. This amendment has been duly made.

Line Number 162 – Sentence Structure:

The sentence structure discussing smoking history at line 162 was revisited and restructured for clarity, and we believe it now conveys the intended message more accurately.

Data Duplication in Table 3 and Lines 166-168:

We thank you for catching the partial data duplication. This has been rectified by eliminating redundancy while maintaining the integrity and clarity of the data presented.

Limitations of the Study:

We acknowledge the omission and have incorporated a section detailing the limitations of the current study, providing a balanced view and acknowledging the constraints encountered during our research.

Conclusion and Its Consistency with Results and Discussion:

You’ve pinpointed a crucial aspect, and in response, we've revisited our conclusion, results, and discussion sections to ensure consistency and coherent representation of aspects highlighted throughout. The conclusion now closely reflects the data and discussions presented in the preceding sections. The enlightening feedback you provided has significantly elevated our manuscript, and for that, we are sincerely grateful. All revisions have been highlighted in the attached manuscript for your convenience and easy reference. We ardently hope that the revisions made, in response to your astute observations, align with your expectations and bring our manuscript a step closer to being worthy of publication.

---

## [Decision Letter · Decision Letter 1]

2 Nov 2023

A SURVEY ON AWARENESS OF THE DISEASE AND PULMONARY REHABILITATION IN BRONCHIAL ASTHMA PATIENTS IN THE UNITED ARAB EMIRATES

PONE-D-23-26953R1

Dear Dr. Bairapareddy,

We’re pleased to inform you that your manuscript has been judged scientifically suitable for publication and will be formally accepted for publication once it meets all outstanding technical requirements.

Kind regards,

Sampath Kumar Amaravadi, Ph.D

Academic Editor

PLOS ONE

Additional Editor Comments (optional):

Reviewers' comments:

Reviewer's Responses to Questions

**Comments to the Author**

1. If the authors have adequately addressed your comments raised in a previous round of review and you feel that this manuscript is now acceptable for publication, you may indicate that here to bypass the “Comments to the Author” section, enter your conflict of interest statement in the “Confidential to Editor” section, and submit your "Accept" recommendation.

Reviewer #1: All comments have been addressed

Reviewer #2: All comments have been addressed

2. Is the manuscript technically sound, and do the data support the conclusions?

Reviewer #1: Yes

Reviewer #2: Yes

3. Has the statistical analysis been performed appropriately and rigorously? 

Reviewer #1: Yes

Reviewer #2: Yes

4. Have the authors made all data underlying the findings in their manuscript fully available?

Reviewer #1: Yes

Reviewer #2: Yes

5. Is the manuscript presented in an intelligible fashion and written in standard English?

Reviewer #1: Yes

Reviewer #2: Yes

6. Review Comments to the Author

Reviewer #1: The authors have diligently and effectively addressed the queries raised, resulting in a noticeable enhancement in the overall quality of the manuscript. As a result, I am pleased to recommend that the manuscript be accepted in its current form.

Reviewer #2: Authors have attended the suggestions well. I recommend the manuscript for publication. I wish the best to the authors.

7. PLOS authors have the option to publish the peer review history of their article (what does this mean?). If published, this will include your full peer review and any attached files.

Reviewer #1: No

Reviewer #2: No

---

## [Editor Report · Acceptance letter]

22 Nov 2023

PONE-D-23-26953R1 

A SURVEY ON AWARENESS OF THE DISEASE AND PULMONARY REHABILITATION IN BRONCHIAL ASTHMA PATIENTS IN THE UNITED ARAB EMIRATES 

Dear Dr. Bairapareddy:

I'm pleased to inform you that your manuscript has been deemed suitable for publication in PLOS ONE. Congratulations! Your manuscript is now with our production department. 

Kind regards, 

on behalf of

Dr Sampath Kumar Amaravadi 

Academic Editor

PLOS ONE